# Climate Change as an Involuntary Exposure: A Comparative Risk Perception Study from Six Countries across the Global Development Gradient

**DOI:** 10.3390/ijerph17061894

**Published:** 2020-03-14

**Authors:** Meredith Gartin, Kelli L. Larson, Alexandra Brewis, Rhian Stotts, Amber Wutich, Dave White, Margaret du Bray

**Affiliations:** 1School of Public Health, University of Alabama Birmingham, 1665 University Blvd., 310F Ryals Public Health Building, Birmingham, AL 35205-0022, USA; 2School of Geographical Sciences and Urban Planning and School of Sustainability, Arizona State University, PO Box 87537-5302, Tempe, AZ 85287, USA; kelli.larson@asu.edu; 3School of Human Evolution and Social Change, Arizona State University, PO Box 872402, Tempe, AZ 85287, USA; alex.brewis@asu.edu (A.B.); rhian.stotts@asu.edu (R.S.); amber.wutich@asu.edu (A.W.); 4School of Community Resources and Development, Arizona State University 411 N. Central Ave., Ste. 550, Phoenix, AZ 85004, USA; dave.white@asu.edu; 5Environmental Studies, Augustana College, 639 38th Street, Rock Island, IL 61201, USA; megdubray@augustana.edu

**Keywords:** risk perceptions, comparative research, climate change, global health

## Abstract

Climate change has been referred to as an involuntary exposure, meaning people do not voluntarily put themselves at risk for climate-related ill health or reduced standard of living. The purpose of this study is to examine people’s risk perceptions and related beliefs regarding (1) the likelihood of different risks occurring at different times and places and (2) collective (government) responsibility and personal efficacy in dealing with climate change, as well as (3) explore the ways in which climate risk may be amplified when posed against individual health and well-being. Previous research on this topic has largely focused on one community or one nation state, and so a unique characteristic of this study is the comparison between six different city (country) sites by their development and national wealth. Here, we collected 401 surveys from Phoenix (USA), Brisbane (Australia), Wellington (New Zealand), Shanghai (China), Viti Levu (Fiji), and Mexico City (Mexico). Results suggest that the hyperopia effect characterized the sample from each study site but was more pronounced in developed sites, suggesting that the more developed sites employ a broader perspective when approaching ways to mitigate their risk against climate-related health and well-being impacts.

## 1. Introduction

Climate change is an environmental hazard that operates at a global scale. Patz and colleagues [1] refer to climate change as an involuntary exposure that threatens to increase worldwide health inequities over time. However, we often examine climate change perception and risk reduction strategies within the boundaries of nation states even though climate change is itself boundless. Being able to explore human experiences with climate change impacts in a comparative way may lead to a broader policy making audience, such as the World Bank, United Nations, or World Health Organization, who may be able to put pressure on nation states where local residents may not have the leverage or agency to do. The purpose of this study is to explore climate change risk perceptions across six different countries as they relate to where climate change is occurring, who should be held responsible when dealing with the impacts of climate change, and what health risks are associated with climate change.

A debate among health professionals centers on whether or not climate change will disproportionately introduce risk to lower-income and more impoverished contexts [2], or whether climate change will create environments for disease emergence or resurgence in both developed and developing countries [1]. Malaria, for example, has shown a striking correlation with poverty [3]; however, recent climate data suggest that the impact of climate change raises the risk for malaria resurgence in the northern hemisphere undergoing major global warming and ecological transformations [4,5,6,7]. This type of ecological change poses a different kind of risk because many countries are not set up or prepared to deal with resurgence of new or endemic infectious diseases.

Additionally, climate change impacts lower-income countries that also have weak or corrupt governments. In Bangladesh, for example, research has shown how health is directly impacted from climate change [8,9,10]. Bangladesh is known as “Ground Zero” for climate change [11]. A recent study found that despite having low educational attainment, rural residents had experience with climate–health-related issues and noted new and emerging diseases as a result of climate variability [8]. This climate–health risk is made worse by a corrupt Bangladeshi government that forestalls any progress to reverse or mitigate climate risk at national level, leaving its population to find ways to cope with climate risks locally and individually [10].

As a result, climate change is considered an involuntary exposure and risk ecology for human health that holds no geographic boundaries. Still, governments should be set up to regulate within their bounded geographies and create policies that can mitigate the exposure rather than exacerbate it. People who live in areas that are rapidly changing due to climatic shifts are also challenged to mitigate their local risks, and how they perceive risk becomes a critical lens through which we can better understand human health and climate change agency at both a local and global scale.

The idea of the hyperopia effect is the phenomenon wherein people tend to view broader-scale risks as more threatening than relatively local ones and may characterize risk perception of climate change in interesting ways [12,13,14]. For example, Leiserowtiz [13] found that people were more likely to perceive risks at a global level, rather than a personal or local level. Specifically, Leiserowtiz [13] found that water shortages were perceived as most likely to occur worldwide over the next 50 years by US residents, followed by increasing rates of diseases, and then decreasing standards of living. This is consistent with other studies identifying the hyperopia effect in the United States [13,15] and the UK [16], but more research is needed to determine whether this effect is also common in developing areas of the world because it can inform much of our international thinking and global policies on global health and climate change-related issues.

Exploring perceived risks among local populations is one way to examine what motivates people to policy action and individual behavior change. Several studies have examined perceptions of climate change risks within country specific impacts [13,17,18]. To contribute to current scholarship on global variation of climate change perception, this study adopts an exploratory, comparative approach by examining responses to multiple survey questions derived from previous work to understand diverse people’s risk perceptions and related beliefs regarding (1) the likelihood of different risks occurring at different times and places and (2) collective (government) responsibility and personal efficacy in dealing with climate change, as well as (3) explore the ways in which climate risk may be amplified when posed against individual health and well-being. The results of this study will provide some insight into idea of the hyperopia effect and how people amplify risk perception around their own individual (internal) risk and global (external) risk.

Previous research has examined perceptions of climate risks in either the developed or developing context to understand the perceptions on the effects of climate change on health in communities [15,19,20,21,22,23]. However, relatively few studies have taken a comparative focus. Most comparative studies have narrowly focused on areas with similar characteristics (e.g., development status or location), as with a study across the United States and the United Kingdom [24]. Although some studies have compared perceptions of climate change among countries in Europe [24,25] or Asia [26], relatively few studies have compared national samples [18]. Moreover, cross-national studies have focused on general national sample in aggregate. This study builds upon this work by examining residents’ perceptions about climate change across six targeted locations, with three each in higher-income/developed and middle-income/developing country sites around the world.

Comparing sites based on national development status can be a useful way to understand how differences in economic status and well-being might influence risk perceptions around climate change and health. Previous research has found that people living under worse socioeconomic conditions are more vulnerable to losses or impacts due to riskier living conditions (e.g., lack of shelter or other resources) and weaker capacities to cope with or mitigate the impacts [27,28]. As a result, residents of lower-income areas often perceive climate change as a greater and more immediate threat than compared to those in wealthier locations [17,18,29]. In turn, lower-resourced or lower-income communities may also be more likely to take direct actions to mitigate or adapt to climate change. The results of this study will provide some insight into how widespread the hyperopia effect is across countries with varying degrees of national GDP and the level to which their populations amplify risk perception around their own individual (internal) risk and global (external) risk.

## 2. Materials and Methods

The data used for this research were collected as part of the 2012 Global Ethnohydrology Study (GES), a multi-year, multi-site study assessing cross-cultural understandings of water and climate change issues [30,31]. The 2012 iteration of the Global Ethnohydrology Study focused on risk perception, responsibility, and climate change. Drawing upon the close-ended survey questions developed by Leiserowitz [13], Spence and colleagues [32], and the World Bank [18], this study provides a comparative analysis of six different sites where authors have on-going ethnographic research. Study participants were selected using a non-probabilistic, purposive sampling strategy in geographically defined local settings, and all respondents had to have lived in the area for at least six months. The survey was conducted in person by trained undergraduate and graduate students.

Responses from a total of 401 participants were analyzed—210 from developed sites and 191 from developing sites. The sample sizes for each site were: Brisbane (72); Wellington (74); Phoenix (64); Shanghai (55); Viti Levu (80); Mexico City (56). Six sites were chosen for this study on the basis of (a) their classification as developed/developing sites, (b) the expectation that each site would represent a unique configuration of climate change risk based upon their economy and ecology and (c) long-standing ethnographic and collaborative research connections in these regions. All subjects gave their informed consent for inclusion before they participated in the study. The study and the protocol were approved by the Ethics Committee of Arizona State University (Project ID: 0804002902).

### 2.1. Study Site Descriptions

The primary objective of this multi-site, comparative study seeks to establish whether or not there are significant differences between the ways in which developed countries and developing countries perceive climatic impacts at both local and global scales. Sites include a mix of higher-income/developed countries and middle-income/developing counties. Phoenix (United States), Brisbane (Australia), and Wellington (New Zealand) are classified as developed/advanced economy sites; and Shanghai (China), Viti Levu (Fiji), and Mexico City (Mexico) are classified as developing/emerging economy sites, based on country-level classifications from the United Nations and International Monetary Fund [33,34]. Furthermore, each site represents a unique configuration of climate change risk based upon their economy and ecology and long-standing collaborative and ethnographic research connections with the authors in these regions.

Brisbane, Australia, is a small river city with a humid subtropical climate and a large suburban population. As the capital city of the state of Queensland, the economy in Brisbane is largely dependent on white-collar work and tourism. Climate change predictions for Brisbane include warming temperatures, greater rainfall variability, more flooding, and more intense cyclones [35]; some of these effects may already be playing out. By 2012, when data were collected, the city had experienced a decade-long drought with significant water restrictions, which were broken the year prior by heavy storms that caused massive flooding. In 2009, the area also experienced a major cyclone and an unusually extreme dust-storm. Major environmental concerns in Australia tend to focus on conservation, especially concerning drought and water resources. Major climate change-related health concerns in Brisbane tend to focus on heat-related illness, infectious disease threats (e.g., dengue), air pollution and respiratory illness, food and water security, and mental illness and stress [36]. Data were collected in and around the city center, where many people come from the suburbs for recreation and shopping.

Wellington, the capital of New Zealand, is the second-most populous urban area in the country and has an economy based on jobs in government agencies, media, the arts, and tourism. While Wellington itself is not prone to regular droughts, much of the North Island, in which Wellington is located, regularly experiences drought. Projected climate change impacts include warming temperatures (with fewer frost days) and some increase in wind speeds [37]. In 2011, the year before we collected data in Wellington, there were two snowfalls at sea level, which is historically uncommon. There is also concern with regards to sea-level rise, coastal erosion, and the potential impacts of coastal storms that are predicted to increase in both frequency and intensity. Predicted health impacts of climate change include injury, infectious disease threats, food and water security, respiratory illness, and mental illness and stress [38]. Data were collected in the city center, which, like Brisbane, draws people in from the suburbs for social, leisure and shopping activities.

As the capital of the southwestern state of Arizona (United States), Phoenix is a large city located in the Sonoran Desert. Phoenix is a fast-growing city known for its rapid population growth, which fuels a growing economy largely based on land development. Although mild winters draw residents and tourists to the region, heat is a major challenge, since summer temperatures can reach higher than 46 °C. Climate change is expected to make the region warmer and drier [39], with increasing health concerns focused on health-related illness, cardiovascular and respiratory illnesses and infectious disease risks associated with mosquito-borne illness [40]. Data were collected in suburban Phoenix neighborhoods.

As China’s largest city, Shanghai is a major financial and transportation hub. Although affluent, Shanghai draws migrants from the countryside and has significant income differentials. Located on the Yangtze River Delta and facing the East China Sea, the low-lying city is vulnerable to periodic typhoons. The city is also at high risk for future sea-level rise and flooding due to climate change and is also predicted to see overall increases in temperature [41,42]. Although pollution is lower compared to other Chinese cities, water and air pollution, as well as industrial waste, are extremely high by world standards. Meanwhile, mosquito-borne and water-borne disease rates are generally low. Public health concerns tied to climate change are focused on heat-related illnesses, fresh water access, and infectious diseases (e.g., Kawasaki) [43]. Nationally, the public’s willingness to engage in critical conversations about environmental issues is shaped by government censorship and low tolerance for direct political action. We collected data in the densely populated urban residential zone of Wujiaochang, northeast of the city center.

Viti Levu, Fiji is the largest and most populous island in the Republic of Fiji. The coasts of Fiji, like most in the island Pacific, are considered particularly vulnerable to sea-level rise, with predictions of rising temperatures with more hot days, increases in wet-season rainfall, and more extreme rainfall days [44]. Coastal Fiji is also susceptible to cyclones; currently, severe cyclones occur every several years. In early 2012, five months before we collected data, severe flooding occurred across Viti Levu, with massive damage to infrastructure and crops, in addition to cases of leptospirosis and typhoid. While malaria free, dengue is considered endemic and outbreaks happen every several years. Climate change refugee-ism is commonly discussed in the Pacific islands, although mainly in regard to lower lying atolls. Data were collected in a coastal peri-rural village on land owned by local indigenous Fijian families—most of whom farm and fish a little but are otherwise engaged as wage earners or entrepreneurs in the cash economy.

Data in Mexico were collected in San Juan de Teotihuacan, a local municipality on the outskirts of Mexico City, approximately 45 km from the city center. The town of less than 50,000 people derives some income from the local tourist attractions at Teotihuacan, and some from seasonal agriculture dependent on rainfall. The region is vulnerable to serious heat waves, droughts, and flooding [45]. The climate is predicted to get hotter and drier as a result of climate change [46]. After the water supply infrastructure was damaged by the 1985 earthquake, water-borne diseases (including cholera) increased throughout the 1990s in Mexico City broadly.

Table 1 reports characteristics of each country, including a summary of their current climate state, and each country has dealt with major events in the recent year(s) prior to the timing of data collection. The first three study sites (above the dashed line) were classified as “developed”; the last three represent “developing” sites. GDP is Gross Domestic Product at the national level; poverty and life expectancy are also national rates. However, the climate information is for the specific regions and cities sampled within nations.

### 2.2. Survey Questionnaire

Perceptions of climate change risks were evaluated from an index developed by Leiserowitz [13]. The first set of questions ask how likely specific events (e.g., water shortages, disease spread, and reduced standard of living) will occur over the next 50 years at both the individual and global level using a 4-point Likert scale from not likely to very likely. This set of questions explicitly examine climatic events to scale. Additional questions addressed the perceptions of current versus long-term (or non-existent) nature of impacts, respondents answered the question: “When do you think climate change will start to substantially harm people in your country?” with a scale ranging from people are currently being harmed, to people will be harmed in 10 years, 25 years, 50 years, 100 years, or never will be harmed.

Additional survey questions were taken from a World Bank [18] survey. Perceived responsibility for climate change action was examined with four questions. Collective responsibility was gauged by: (1) “Do you think your country does or does not have a responsibility to take steps to deal with climate change?” (with survey participants responding in the affirmative or negative); and, (2) “To deal with the problem of climate change, do you think your government is doing—too much, not enough, or about the right amount?” To gauge people’s sense of personal responsibility, a 5-point agree/disagree scale from Spence and colleagues [32] was used, specifically for: (3) “I can personally help to reduce climate change by changing my behaviors.” and (4) “I personally feel that I can make a difference with regard to climate change.” Again, the scale is determined by the actor who is considered responsible (i.e., individual versus government) and by the action or impact as being perceived to be enough to buffer or mitigate climate risks and events.

### 2.3. Data Analysis

Data were analyzed using SPSS version 25.0 (IBM, Armonk, NY, USA). Several statistical tests were employed to explore differences between the developing and developed sites on climate and risk perception. Descriptive statistics reveal overall trends in the level of risk perceptions for the entire sample. Mann–Whitney U tests were employed with the ordinal data and chi-square tests were used on categorical data to identify statistically significant differences between residents from developing versus developed sites. ANOVA tests were also conducted to identify the site-level distinctions in risk perceptions; Tukey’s post-hoc tests revealed which pair-wise differences were statistically significant.

## 3. Results

Table 2 and Table 3 report the descriptive statistics for the total sample on key dependent variables that define the various kinds of risk perceptions outlined by the survey questionnaire.

Table 4 reports test statistics for all group comparisons between the variables. All comparisons between the developed and developing countries were found significant, except “government responsibility” where very little variation was found between the sites such that people overwhelmingly attribute responsibility to their national government.

### 3.1. Climate Change Impacts by Scale

The items of the risk perception scale [9] rank in an interesting way; disease spread was the greatest concern, followed by reduced standard of living and water shortages. In each item, the primary concern was at the individual level and then the global level (Table 4).

Post-hoc tests identified some overlap between countries individually that do not fall into the defined developed/less developed context (Table 5). China and Fiji (as developing countries) appear to align with all developed countries (Australia, New Zealand, and the United States) on the disease scale items (both individual and global levels). However, the United States drops out of significance with these countries for perceived impacts on standard of living and water shortages (both individual and global levels). This is likely because New Zealand, Australia, and Fiji all share similar sea-level and climate risks that has created many climate refuges from the Pacific islands who migrate to Australia, New Zealand, and other parts of Asia, including China.

Meanwhile, Mexico is an outlier in almost all items on the scale (Table 6). Post-hoc analyses found that it almost exclusively aligns in its own subgroup and context from all other sites. Homogeneous subsets from Tukey HSD post-hoc analysis found Mexico as its own separate subset from other countries on three major items: (1) standard of living worldwide, (2) “my” standard of living, and (3) water shortages “where I live”. For the other items, Mexico was in a subset with China on the item water shortages worldwide; Mexico was found in a subset with Fiji on the item, “my chances of” disease; and Mexico was found in a subset with Fiji and China on the item, diseases worldwide.

### 3.2. Location and Timing of Impacts

We evaluated perception around when people think climate change will cause harm and whether or not climate change events will be worse in poorer versus wealthy countries, or both but in different ways. Results identified that our developing sites perceive climate change to be causing harm currently or within the next 10 years, whereas most in the developed countries perceive the risk to be more than 25 years from the time of survey (Table 7).

To address climate change impacts to a location and scale across country income/wealth gradient, there is very little variation between developed and developing countries (Table 4). People across development contexts perceive that harm will occur in both poorer and wealthier countries with different impacts between them. However, Australia and approximately one-third of the population in New Zealand believe that climatic events will be more harmful to poorer countries (Table 8); this is likely due to the shared experience of climate refugees from the pacific islands, which are often poorer countries, being displaced.

### 3.3. Collective and Individual Responsibility

No significant differences were found between developed and developing countries when asked about their governmental responsibility; rather, the majority of each site in the sample (Table 9) agree that their governments do have responsibility. However, some significant difference existed between developed and developing countries with respect to whether or not their government is doing enough (Table 4). In general, the majority do not consider that the government is doing enough (Table 10). Approximately 30% of the respondents from developed countries believe that their governments are doing about the right amount, compared to less than half this portion in developing countries.

Personal responsibility was assessed by asking whether respondents feel they can make a difference (Table 11) and whether they can help by changing their behavior (Table 12). In both cases, the developing countries are almost split between agreeing or feeling neutral about their personal responsibility or ability to impact (or mitigate) climate change, whereas the developing countries mostly agree. Interestingly, approximately one-third of the Fijian sample believe that changing their behavior can have an impact, while a little more than one-quarter disagree (Table 12).

## 4. Discussion

Disease risk, as associated with climate change, is found to be the highest-ranked climatic impact, followed by reduced standard of living and water shortages. For each climatic impact, perceptions are at the individual level first and then the global level. Research suggests that amplified perceptions of specific risks may be partially influenced by exposure to these risks, similar to the familiarity hypothesis [53,54,55] and in accordance with the finding [41] that people who are directly exposed to a particular risk are more likely to perceive future threats. In particular, people in Mexico City and Viti Levu—which have experienced climate-sensitive diseases such as dengue—express the strongest perceived likelihood of a change in contracting diseases due to climate change, suggesting that experience with disease amplifies risk perceptions.

Some research suggests that familiar risks are seen as less threatening than unknown or unfamiliar risks [53,54,55]. This might be due to the presence of economic benefits that outweigh potential risks [54,55], a sense of pride that makes members of the local community unwilling to admit to risks or concerns [56], or higher feelings of control over local and familiar risks [12,57]. However, other studies show that personal experiences with risky situations heighten perceptions and concerns about associated impacts [32,57,58]. The discrepancy here might be due to the nature of the risk involved. For example, perceiving a warming climate might be hampered by the relatively slow and incremental onset of the resulting impacts, which may in turn garner less concern compared to acute risks and short-term impacts [14]. On the other hand, personally poignant risks, such as local air and water pollution, can be directly experienced, and thus, might be identified as more likely or concerning [58]. More research is needed along this line of inquiry.

The patterns are less clear for living standards. In Viti Levu, for example, people live in the poorest region and with the lowest life expectancy when compared to the other countries in this study; however, our sample does not have particularly heightened concerns about living standards. Respondents in Mexico City—where poverty and inequities are greater—do exhibit amplified perceptions regarding the impacts of climate change on their standard of living. This finding seems to suggest that sites with high economic inequality (such as a municipality in Mexico) might be more likely to expect reduced living standards than those with lower economic inequality (such as an indigenous village in Fiji); and that the ‘developed’ status of the country is not the binding factor. Instead, relative economic inequality is a more important factor that binds the lived experience and merits further investigation in terms of its role in shaping perceptions and actions regarding climate change and health.

Although results for the United States were comparable to our results for Phoenix, the World Bank’s findings for China and Mexico are quite different from ours [18]. The World Bank study [18] surveyed residents of fifteen countries to find that people believe both wealthy and poor countries will experience the impacts of climate change equally, despite the general view that poor regions will be disproportionately affected [59]. Our study found that respondents believe both poor and rich countries will be affected but in different ways, and only Australia lean towards the belief that poorer regions will be disproportionately affected. In the original study by the World Bank, mostly developing nations (specifically, Senegal, Bangladesh, and Turkey) perceive greater harm to lower-income nations compared to higher-income countries. These results underscore the importance of analyzing localized impacts and perceptions, as opposed to the more common national surveys because variation within countries do exist and can amplify the relationship between risk and inequity.

Approximately 83% of survey respondents from another Mexico-based study [18] perceive current harm from climate change versus 69% of our respondents from Mexico City (Table 7). The discrepancies in results between the two studies were even more dramatic for China. While only 35% of our study respondents in Shanghai indicate current harm from climate change (Table 7), 71% of Chinese respondents indicate that people are being harmed currently [18]. This could be attributed to the fact that our study examines perceptions of city dwellers, who may feel more buffered by the impacts of climate change. Nevertheless, further research might examine the influences of national versus (local) site level, contextual factors on people’s perceptions of climate change.

Despite feeling more vulnerable to risks, our sample exhibits stronger feelings of efficacy in their ability to take actions that will help mitigate climate change. This might be due to their personal experiences confronting the impacts of climate change, which Spence and colleagues [32] find also increase people’s perceptions of their own abilities to handle a risk. This finding is troubling, however, since developed nations have been, by far, the largest contributors to greenhouse gas emissions, and thus their citizens should be taking action to reduce their impacts. Especially in the United States, which is a top greenhouse gas emitter, low feelings of efficacy may constrain changes in personal behaviors that can help mitigate climate change. Even in China, the largest contributor to greenhouse gas emissions worldwide [60], a majority of residents in the study site (64%) exhibit higher efficacy to reduce climate change by changing behaviors compared to those in the United States (41%).

In contrast to feelings of personal efficacy, our sample generally agrees that the government has a responsibility to deal with climate change. This is the only variable that did not differ based on development status and is consistent with the World Bank study [18] results. Interestingly, this sentiment was highest in Shanghai, perhaps due to its top–down government structure, and lowest in Viti Levu, perhaps due to a sense of relative influence in a smaller-scale, representative political system where indigenous Fijian values and rights are given weight. Although not the focus of this study, it would be important to also explore the responsibility of governments to provide health care relief for climatic-related disease and well-being. Research indicates that most governments, where climate change is transforming local ecological risk, are not ready for major vector-disease outbreaks or resurgence of infectious disease [4,5,6,7].

## 5. Conclusions

Climate change is considered an involuntary exposure that operates at a global scale. As an involuntary exposure, climate change is capable of massive destruction to local ecosystems and local social, economic, and health systems. The results of this study demonstrate that people, regardless of their own wealth or their country’s wealth and economic security, are concerned with their health first and their standard of living second. Their concern is not just about their own livelihoods in their local contexts but also about the global community. The hyperopia effect may be a way to further examine how the exposure to climate change on a global scale is perceived in terms of risks that are familiar and disconnected from local contexts. Our findings suggest that the hyperopia effect [12,14] characterized the sample from each study site but was more pronounced in developed sites, suggesting that the more developed sites employ a broader perspective when approaching ways to mitigate their risk against climate-related health and well-being impacts. The hyperopia effect offers an interesting lens through which we can examine global health and climate change.

This study did not examine how climate change risks are managed specifically within the study sites. However, this study did find that people do consider ways to manage and mitigate climate-related risk, particularly as associated with threats to their health and well-being. Residents of developing sites see the impacts of climate change as more likely—especially at the proximal, local scale and near-term timeframes—compared to residents of developed ones. Not only do they feel more vulnerable in these ways, but they rank their health and livelihood above all other impacts and feel a stronger sense of efficacy regarding their personal abilities to mitigate climate change. At the same time, residents from all sites overwhelmingly expressed their belief that the government has a responsibility to address climate change. More research should be directed to examine the direct links between what people perceive and how their behaviors are informed by those perceptions to mitigate disease risk from climate-related impacts.

Although this study examines the differences and similarities between developed and developing economies, a key finding in this study points to an opportunity to reframe the discussion. We often talk about countries in terms of developed versus developing, rich or poor, us and them, whereas the data in this study did not clearly sort into those categories, particularly when examining the impact of climate change on economic livelihood. Data in this study indicate that income inequity within countries regardless of their global development status may be a strong factor framing the real-world experience with climate risk. In other words, people who have to overcome great economic inequity may share some kind of global climate, involuntary exposure and could be a new lens through which we examine the ways in which global experiences are shared transnationally and cross-culturally.

Global health is a lens through which we can examine the ways in which global and local forces converge to create health risks for populations. The global forces are often removed from direct links to communities, which is why the hyperopia effect is so useful in studying global health threats, like climate change and vulnerable populations. Global health also provides a means through which we can reframe the discussion, as the field considers that disease holds no boundaries and health should be explored transnationally [61]. The best example of this perspective from the current study is through the shared risk perceptions among the sites located in the South Pacific, in Australia, Fiji, and New Zealand. The South Pacific is the region where the first groups of people were displaced due to climate change as refugees in political discourse [62] and the South Pacific is considered on the front lines for sea-level rise and climatic threats to human survival [63,64]. Climate change operates at a global scale, with very local impacts. Further, in the most extreme cases, climate change will lead to international migration, which is why comparative and cross-cultural perspectives are needed in research, particularly with the increasing population of climate refugees.

Ultimately, the major takeaway from this study is that more research needs to be carried out to evaluate risk perceptions around real climate–health impacts on local populations in diverse places. What kinds of health issues are riskier in various environments and how are those specifically experienced by local residents, governmental agencies, and health workers should become the line of inquiry for future research and interventions. Health care workers are as much on the front lines of climate–health impacts as those who suffer disease from climate change, and they are also exposed to the same climate-related risks. More research is needed to address the needs of health systems to policy makers, and so the mitigation of risk is not the burden of an individual but the responsibility of the government for its public’s health. Research should be designed to assess what people know about climate-related health risks and what is needed from their health systems, especially considering a range of development indicators (i.e., government corruption or coherency, income and gender inequities).

## Figures and Tables

**Table 1 ijerph-17-01894-t001:** Characteristics of study sites.

Sites (Nation)	GDP Per Capita	Poverty Rates	Life Expectancy	Climate Type	Annual Precipitation (mm)	Anticipated Climate Changes
Brisbane (Australia)	$67,036	13%	82	Humid subtropical	1148.8 [47]	Warmer, drier, increased flooding and cyclone intensity [35]
Wellington (New Zealand)	$37,749	15%	81	Temperate marine	957.0 [48]	Warmer, wetter, increased westerly winds [37]
Phoenix (United States)	$49,965	15%	79	Semi-arid desert	210.8 [49]	Warmer, drier, increased drought [39]
Shanghai (China)	$6188	13%	75	Humid subtropical	1173.4 [50]	Warmer, wetter [41], increased flooding, sea-level rise [42]
Viti Levu (Fiji)	$4438	31%	70	Tropical marine	1800.0 [51]	Warmer, wetter [44]
Mexico City (Mexico)	$9747	51%	77	Temperate semi-arid	709.0 [52]	Warmer, drier [46], increased drought and flooding [45]

**Table 2 ijerph-17-01894-t002:** Descriptive statistics for ordinal variables.

Ordinal Variables	Mean	Median	St. Dev.	Response Range
Water shortages worldwide ^1^	3.19	3.0	0.926	1–4
Water shortages “where I live” ^1^	2.81	3.0	1.023	1–4
Diseases worldwide ^1^	2.99	3.0	0.936	1–4
“My chances of” disease ^1^	2.60	3.0	1.049	1–4
Standard of living worldwide ^1^	2.87	3.0	0.936	1–4
“My” standard of living ^1^	2.52	3.0	1.014	1–4
Timing of local harm ^2^	2.68	2.0	1.568	1–6
Personal ability to reduce effects ^3^	3.63	4.0	1.168	1–5
Personal ability to make a difference ^3^	3.56	4.0	1.065	1–5

Notes: responses ranged from ^1^ 1 = not at all to 4 = very likely; ^2^ 1 = people are being harmed now to 6 = people will never be harmed; and ^3^ 1 = disagree strongly to 5 agree strongly.

**Table 3 ijerph-17-01894-t003:** Frequencies for nominal variables.

Categorical Variables	Freq.	Percent
*Climate change will be:*		
More harmful to wealthy countries	11	2.8%
More harmful to poor countries	90	22.8%
Equally harmful to wealthy/poor countries	74	18.8%
Both will be affected, but in different ways	219	55.6%
*Your country dealing with climate change:*		
Does have a responsibility	353	89%
Does not have a responsibility	43	11%
*Your country is doing:*		
Too much	25	6.3%
Not enough	288	72.3%
About the right amount	85	21.4%
*When do you think climate change will start to harm people in your country?*		
People are being harmed now	135	34%
In 10 years	62	16%
In 25 years	72	18%
In 50 years	62	16%
In 100 years	45	11%
Never	18	5%

**Table 4 ijerph-17-01894-t004:** Statistical tests of differences based on development status.

Individual Variables	Test Statistic	*p* Value
*Climate Change Impacts by Scale*		
“My chances of” diseases (individual) ^1^	28,458.0	<0.001
Diseases worldwide (global) ^1^	27,142.0	<0.001
“My” standard of living (individual) ^1^	26,015.5	<0.001
Standard of living worldwide (global) ^1^	25,862.5	<0.001
Water shortages “where I live” (individual) ^1^	25,775.0	<0.001
Water shortages worldwide (global) ^1^	22,844.0	<0.001
*Location and Timing of Impacts*		
Timing of local harm ^1^	9291.5	<0.001
Affected countries ^2^	30.7	<0.001
*Collective and Individual Responsibility*		
Government responsibility ^2^	0.029	0.866
Government effectiveness ^2^	21.1	<0.001
Personal ability to reduce effects ^1^	22,217.0	0.031
Personal ability to make a difference ^1^	24,218.0	<0.001

Notes: ^1^ Mann–Whitney U test. ^2^ Chi-square test.

**Table 5 ijerph-17-01894-t005:** Tukey HSD post-hoc analysis comparing China and Fiji to developed countries.

Scale Item	Pairs (I, J)	Mean Difference (I–J)	Significance
“My chances of” diseases	China, Australia	0.75654	0.000
	China, New Zealand	0.96135	0.000
	China, United States	0.54514	0.016
	Fiji, Australia	0.90661	0.000
	Fiji, New Zealand	1.11142	0.000
	Fiji, United States	0.69521	0.000
Diseases worldwide	China, Australia	0.70635	0.000
	China, New Zealand	0.65806	0.000
	China, United States	0.50000	0.022
	Fiji, Australia	0.67216	0.000
	Fiji, New Zealand	0.62387	0.000
	Fiji, United States	0.46581	0.018
“My” standard of living	China, Australia	0.50396	0.038
	China, New Zealand	0.76442	0.000
	Fiji, Australia	0.44760	0.044
	Fiji, New Zealand	0.70806	0.000
Standard of living worldwide	Fiji, Australia	0.58360	0.001
	Fiji, New Zealand	0.41250	0.001
Water shortages “where I live”	China, New Zealand	0.83636	0.000
	Fiji, New Zealand	1.03750	0.000
Water shortages worldwide	see Table 6		

**Table 6 ijerph-17-01894-t006:** Tukey HSD post-hoc analysis comparing Mexico to full sample sites.

Scale Item	Australia	China	Fiji	New Zealand	United States
“My chances of” diseases	1.30401 **	0.54747 *	0.3974	1.50883 **	1.09261 **
Diseases worldwide	1.08895 **	0.38269	0.41679	1.04066 **	0.88260 **
“My” standard of living	1.07721 **	0.57324 *	0.62960 *	1.33766 **	0.94972 **
Standard of living worldwide	1.04286 **	0.59091 **	0.46250 *	0.87500 **	0.73438 **
Water shortages “where I live”	0.72078 **	0.72727 **	0.52614 *	1.56364 **	0.82926 **
Water shortages worldwide	0.85882 **	0.42264	0.70789 **	0.71549 **	0.80000 **

Note: Shown in the cell is (Mean Difference = Mexico—Country); * *p* < 0.05, ** *p* < 0.001.

**Table 7 ijerph-17-01894-t007:** Frequency on item: when do you think climate change will start to substantially harm people in your country?

Sample	People are Harmed Now	In 10 Years	In 25 Years	In 50 Years	In 100 Years	Never
Developed Countries	17.6% (36)	9.8% (20)	25.4% (52)	20% (41)	19.5% (40)	7.8% (16)
Australia	9.9% (7)	11.3% (8)	23.9% (17)	26.8% (19)	21.1% (15)	7% (5)
New Zealand	12.9% (9)	12.9% (9)	28.6% (20)	21.4% (15)	14.3% (10)	10% (7)
United States	31.3% (20)	4.7% (3)	23.4% (15)	10.9% (7)	23.4% (15)	6.3% (4)
Developing Countries	52.4% (99)	22.2% (42)	10.6% (20)	11.1% (21)	2.6% (5)	1.1% (2)
China	35.2% (19)	11.1% (6)	14.8% (8)	27.8% (15)	9.3% (5)	1.9% (1)
Fiji	52.5% (42)	31.3% (25)	10% (8)	5% (4)	0% (0)	1.3% (1)
Mexico	69.1% (38)	20% (11)	7.3% (4)	3.6% (2)	0% (0)	0% (0)
Full Sample	34.3% (135)	15.7% (62)	18.3% (72)	15.7% (62)	11.4% (45)	4.6% (18)

**Table 8 ijerph-17-01894-t008:** Frequency on item: climate change will be more harmful in which country groups.

Sample	Wealthy Countries	Poorer Countries	Equally Harmful to Both	Both Affected Differently
Developed Countries	2.9% (6)	33.7% (69)	18.5% (38)	44.9% (92)
Australia	4.3% (3)	40% (28)	22.9% (16)	32.9% (23)
New Zealand	4.1% (3)	35.6% (26)	8.2% (6)	52.1% (38)
United States	0% (0)	24.2% (15)	25.8% (16)	50% (31)
Developing Countries	2.6% (5)	11.1% (21)	19% (36)	67.2% (127)
China	0% (0)	22.2% (12)	20.4% (11)	57.4% (31)
Fiji	6.3% (5)	5.1% (4)	15.2% (12)	73.4% (58)
Mexico	0% (0)	8.9% (5)	23.2% (13)	67.9% (38)
Full Sample	2.8% (11)	22.8% (90)	18.8% (74)	55.6% (219)

**Table 9 ijerph-17-01894-t009:** Frequency of government responsibility.

Sample	Government Has Responsibility	Government Does Not Have Responsibility
Developed Countries	88.9% (184)	11.1% (23)
Australia	87.3% (62)	12.7% (9)
New Zealand	87.7% (64)	12.3% (9)
United States	92.1% (58)	7.9% (5)
Developing Countries	89.4% (169)	10.6% (20)
China	98.2% (54)	1.8% (1)
Fiji	81% (64)	19% (15)
Mexico	92.7% (51)	7.3% (4)
Full Sample	89.1% (353)	10.9% (43)

**Table 10 ijerph-17-01894-t010:** Frequency of government effectiveness.

Sample	Government Does Too Much	Government Does Not Do Enough	Government Does About The Right Amount
Developed Countries	4.8% (10)	64.9% (135)	30.3% (63)
Australia	6.9% (5)	59.7% (43)	33.3% (24)
New Zealand	2.7% (2)	56.2% (41)	41.1% (30)
United States	4.8% (3)	81% (51)	14.3% (9)
Developing Countries	7.9% (15)	80.5% (153)	11.6% (22)
China	5.5% (3)	90.9% (50)	3.6% (2)
Fiji	15% (12)	65% (52)	20% (16)
Mexico	0% (0)	92.7% (51)	7.3% (4)
Full Sample	6.3% (25)	72.4% (288)	21.4% (85)

**Table 11 ijerph-17-01894-t011:** Frequency of agreement level on item: personally, I feel that I can make a difference with regard to climate change.

Sample	Strongly Disagree	Disagree	Neither Agree/Disagree	Agree	Strongly Agree
Developed Countries	7.3% (15)	12.2% (25)	31.7% (65)	37.1% (76)	11.7% (24)
Australia	5.9% (4)	14.7% (10)	30.9% (21)	35.3% (24)	13.2% (9)
New Zealand	4.1% (3)	13.7% (10)	34.2% (25)	38.4% (28)	9.6% (7)
United States	12.5% (8)	7.8% (5)	29.7% (19)	37.5% (24)	12.5% (8)
Developing Countries	3.2% (6)	9.6% (18)	14.9% (28)	48.9% (92)	23.4% (44)
China	1.9% (1)	14.8% (8)	29.6% (16)	48.1% (26)	5.6% (3)
Fiji	1.3% (1)	10.3% (8)	10.3% (8)	51.3% (40)	26.9% (21)
Mexico	7.1% (4)	3.6% (2)	7.1% (4)	46.4% (26)	35.7% (20)
Full Sample	5.3% (21)	10.9% (43)	23.7% (93)	42.7% (168)	17.3% (68)

**Table 12 ijerph-17-01894-t012:** Frequency on item: personally, I can help reduce climate change by changing my behavior.

Sample	Strongly Disagree	Disagree	Neither Agree/Disagree	Agree	Strongly Agree
Developed Countries	6.3% (13)	10.6% (22)	22.1% (46)	43.3% (90)	17.8% (37)
Australia	7.1% (5)	11.4% (8)	25.7% (18)	37.1% (26)	18.6% (13)
New Zealand	2.7% (2)	13.5% (10)	16.2% (12)	51.4% (38)	16.2% (12)
United States	9.4% (6)	6.3% (4)	25% (16)	40.6% (26)	18.8% (12)
Developing Countries	8.4% (16)	13.1% (25)	6.3% (12)	44% (84)	28.3% (54)
China	0% (0)	7.3% (4)	9.1% (5)	63.6% (35)	20% (11)
Fiji	13.8% (11)	23.8% (19)	6.3% (5)	38.8% (31)	17.5% (14)
Mexico	8.9% (5)	3.6% (2)	3.6% (2)	32.1% (18)	51.8% (29)
Full Sample	7.3% (29)	11.8% (47)	14.5% (58)	43.6% (174)	22.8% (91)

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
