# Peer review of "Climate Change as an Involuntary Exposure: A Comparative Risk Perception Study from Six Countries across the Global Development Gradient"

_ijerph, 2020, doi:10.3390/ijerph17061894_

Round 1
Reviewer 1 Report
The authors examined people's risk perceptions and related beliefs on climate change using six cities from six countries. It's a very novel study which shows that developed cities have broader perception on the risk of climate change.
Major comments:
The whole structure needs to be adjusted. For example:
1).1.1 Study Site Descriptions should be moved to 2.2 Materials and Methods section.
2). Most part in Conclusion section should be moved to Discussion section
Minor comments:
The authors should mention what software or tool they used in their data analysis process.
Author Response
Thank you so much for taking the time to review our article. I have responded to your comments below and I hope you will find the new version to be much more improved.
Reviewer Comment: The authors examined people's risk perceptions and related beliefs on climate change using six cities from six countries. It's a very novel study which shows that developed cities have broader perception on the risk of climate change.
Major comments:
The whole structure needs to be adjusted. For example:
1).1.1 Study Site Descriptions should be moved to 2.2 Materials and Methods section.
2). Most part in Conclusion section should be moved to Discussion section
Author Response: The study site descriptions was moved to the Materials and Methods section. Regarding #2, we agree that the conclusion needed work and ended up moving a good portion of the conclusion into the introduction as it read better as background to the study. The discussion was also edited as well.
Reviewer Comment: Minor comments: The authors should mention what software or tool they used in their data analysis process.
Author Response: We used SPSS v25 and this was added to the methods section.
Reviewer 2 Report
Data was collected in 2013 now 7 years ago and much has changed – the Paris accord, Donald Trump replace Barack Obama, extinction rebellion, etc. I therefore think the introduction and discussion need to explain why such old data is relevant in a fast changing world. I’ve been to Shanghai in the last few years and they had a very strong green ethic and action.
I found the variations among the countries/ regions interesting but they made it hard to see a clear developed vs developing dichotomy especially with only approx. 400 across the whole 6 sites.
Also I am very aware of the local political conflict around climate change in Australia and attitudes in Queensland (although Brisbane is more progressive than rural regions). This local knowledge made me aware of the need to have far more political background and environmental attitudes on each of the 6 regions surveyed. The authors say they are very knowledgeable about each of the areas and it would have been good to have far more of that information in the introduction and discussion. It was interesting to hear about Bangladesh but it would have been far more relevant to hear about the regions surveyed.
Some more specific notes
Wellington is the capital of NZ therefore government is a major activity
Tables 2 and 3 appear to be results so I’m not sure why they are in the analysis section
Table 4 tells us there is a difference but not whether developed or less-developed was higher
The total sample is approx. 400 which is not many over 6 countries. When variables of interest have several categories, eg Table 8, the numbers get very small
Questions with Likert scale type response options were treated as categories rather than calculating means and making stronger statistical comparisons. The distributions for the totals were not bi-modal so means would be appropriate.
Author Response
Thank you so much for the detailed review of our manuscript. We were thrilled to see that the article brought up so many thoughts about the current events and relevancy of our questions. We have made some major modifications to the draft and its structure given many of your comments and suggestions, and we hope you will find this version much improved. Below, I have outlined detailed responses to each of your comments.
Comments and Suggestions for Authors
Comment #1: Data was collected in 2013 now 7 years ago and much has changed – the Paris accord, Donald Trump replace Barack Obama, extinction rebellion, etc. I therefore think the introduction and discussion need to explain why such old data is relevant in a fast changing world. I’ve been to Shanghai in the last few years and they had a very strong green ethic and action.
Limitations and data was collected in 2013 and that this a rapidly changing socio-political landscape…
Response #1: We completely understand the hesitation with the year the data was collected. We added a line for limitations to the study stating, "The data were collected in 2013, and capture a fast-changing and dynamic phenomenon. As such, the findings may not reflect current trends." However, given that there have been major political shifts and changes in global forums, a review of the literature did not find any new studies that report more localized perspectives as our study does. So, even though they may not reflect current events, I think the significant piece is the framework and design of examining cross-cultural and transnational perspectives which this study attempts to do in a very exploratory way.
Comment #2: I found the variations among the countries/ regions interesting but they made it hard to see a clear developed vs developing dichotomy especially with only approx. 400 across the whole 6 sites.
Response #2: There wasn’t a very clear dichotomy and that might not have been very well portrayed in text. Discussion and conclusion have been revised to help show that the dichotomy might not be the best way to draw comparisons and instead we should be challenged to find other factors to categorize and compare across different countries. For example, see revised Conclusion, Paragraph 3 & 4.
Comment #3: Table; GPD and infectious disease rate.
Response #3: I really like the idea of including more information around disease unfortunately, we couldn’t find enough information from each site to build an appropriate table/comparison. We did include a few extra sentences in each site description paragraph that related to common climate-related health issues but not all sites have those reported.
Comment #4: Also I am very aware of the local political conflict around climate change in Australia and attitudes in Queensland (although Brisbane is more progressive than rural regions). This local knowledge made me aware of the need to have far more political background and environmental attitudes on each of the 6 regions surveyed. The authors say they are very knowledgeable about each of the areas and it would have been good to have far more of that information in the introduction and discussion. It was interesting to hear about Bangladesh but it would have been far more relevant to hear about the regions surveyed. More detail about the sites.
Response #4: We added some additional detail in the site descriptions and reframed the discussion and conclusion to better highlight the more site-specific details.
Comment #5: Wellington is the capital of NZ therefore government is a major activity
Response #5: Agreed. That is mentioned in the site description.
Comment #6: Tables 2 and 3 appear to be results so I’m not sure why they are in the analysis section
Response #6: Agreed. We moved them to the results.
Comment #7: Table 4 tells us there is a difference but not whether developed or less-developed was higher
Response #7: Yes that is correct; Table 4 was used to establish if the difference between the two was significant and not which was higher. Table 4 is the overall picture and then within the sections of 3.1-3.3, we provided the detailed breakdown across the sites to show the direction of the statistic.
Comment #8: The total sample is approx. 400 which is not many over 6 countries. When variables of interest have several categories, eg Table 8, the numbers get very small
Response #8: Yes that is correct. It is the reason why we refer to this as an “exploratory” study and focus on the ‘localized’ perspective in our discussion.
Comment #9: Questions with Likert scale type response options were treated as categories rather than calculating means and making stronger statistical comparisons. The distributions for the totals were not bi-modal so means would be appropriate.
Response #9: I agree that this would be an option. We did not take this approach because we were less interested in comparing the indices and more interested in exploring the relationship between the indicators of the scales. The study really was designed to be more exploratory with less rigor in the analysis.
Round 2
Reviewer 1 Report
The revision version was improved a lot, hence much more readable.